# OpenReview forum: "Model-free reinforcement learning with noisy actions for automated experimental control in optics"
_TMLR — Accepted by TMLR_

### Review · Reviewer_4MWG · 2025-02-23

**Summary Of Contributions:**

The paper studies the challenging problem of coupling laser light into an optical fiber within a complex, noisy, and partially observable environment. The authors demonstrate that reinforcement learning (RL) can overcome these challenges by training an agent directly on the physical setup, eliminating the need for an accurate simulation. They evaluate several RL algorithms and show that the experimental results achieve performance comparable to a human expert. Additionally, the paper investigates the benefits of curriculum learning and pretraining on a virtual testbed to enhance training efficiency, highlighting the potential of these techniques for real-world optical control applications.

**Audience:**

Yes

**Broader Impact Concerns:**

No broader impact concerns.

**Claims And Evidence:**

Yes

**Requested Changes:**

1. Discuss the differences between the optical environment in this study and classic noisy RL environments (e.g., robotics control in Mujoco).
2. Expand the discussion on training time and sample efficiency, including strategies to reduce training duration (such as sim-to-real approaches), and evaluate the impact of virtual pretraining.
3.  Discuss the similarities and differences between virtual and real-world environments, focusing on the transferability of reward and observation designs.
4.  Provide a more detailed analysis of the reset strategy, including its implications for both training and real-world control.

Please check more details in the [Weaknesses]  section above.

**Strengths And Weaknesses:**

Strengths
1. Practical Relevance: By addressing an important problem in automating optical experiments, the work has the potential to reduce manual tuning and improve experimental efficiency in optical research.
2. Real-World Experimentation: Training directly on the experimental setup provides robust evidence of the practical viability of RL in complex optical systems.
3. Detailed Analysis: The study offers comprehensive comparisons between different RL algorithms and thoroughly explores the benefits of curriculum learning, enhancing experimental validation.
4. Virtual Testbed Utilization:: It is beneficial to leverage a virtual testbed to inform real-world training strategies, enabling fine-tuning of parameters before lab deployment.

Weaknesses
1. Comparison to Classic Noisy Environments: It is beneficial to clearly differentiate the optical environment in this study from typical noisy RL environment settings, such as those in robotics (e.g., Mujoco).
2. Training Time and Sample Efficiency: Although the results are promising, the long training duration (up to 4 days for high goal power) raises practical concerns. It is better to conduct discussions about potential strategies —such as improved exploration techniques, hybrid approaches, or advanced sim-to-real methods—to reduce training time, and by reducing real-world training time.
3. Real vs. Virtual Environment: It is beneficial to further analyze the transferability of ablation studies from the virtual testbed to real-world experiments. Discussion is needed on how reward and observation designs in virtual testbeds can be transferred to real-world experiments. In addition, it is beneficial to discuss similarities/differences between virtual testbed and real experiment.
4. Reset Procedure Complexity: It would be beneficial to discuss the reset mechanisms in more detail. For instance, does the reset mechanism apply only during real-world training, or does it also extend to virtual training and controlling on real-world systems? A more thorough exploration of reset strategies, including alternative resetting methods, is beneficial.

---

> ### Author Response · Authors · 2025-03-06
>
> Thank you very much for your appreciative feedback. Before making changes to the paper we will wait until all reviewers have replied. Here we would like to follow up on your requested changes:
>
> Regarding 1.: We can add a comparison of our environment and standard robotics environments. However, this is not an easy task as we are not aware of standardized criteria for such comparisons (although the encountered problems can be similar). The obvious differences are that many robotics environments have higher dimensional action and observation spaces but backlash is often not discussed in simulations (as far as we know). In sim-to-sim transfer, it was found that different backlash seems to have a great effect on the transferability of policies, but agents are still able to learn in an environment with backlash (https://proceedings.mlr.press/v87/golemo18a/golemo18a.pdf). Backlash is also what we found to be a challenge for our agent. Would such a discussion be the right direction for you? We intend to include this discussion in our related work section.
>
> Regarding 2.: We are happy to expand the discussion on training time and sample efficiency in the outlook.  Note, however, that our main goal is to show that successful training is possible without relying on a sim-to-real transfer but training solely on the experiment. We discussed pretraining on the virtual testbed in Appendix D.3. Would you prefer this discussion to be shifted to the main text?
>
> Regarding 3.: We can further elaborate onthe similarities and differences between the virtual testbed and the real experiment at the end of Section 4. Please note, however, that the goal was not to make them perfectly comparable but only to have a toy environment to test  strategies.
>
> Regarding 4.: We discuss the impact of different reset methods in Appendix C.3.  Could you let us know what aspects you think are not sufficiently discussed there and whether you think this should appear in the main text and not only in the Appendix?
>
> Thank you very much for your time.

---

> > ### Author Response · Authors · 2025-04-30
> >
> > Thank you for your patience. We have now changed our manuscript incorporating your suggestions and uploaded a new version. Please see our general answer on top for logs of the changes.

---

### Review · Reviewer_sR4c · 2025-03-05

**Summary Of Contributions:**

This paper presents an application of model-free reinforcement learning algorithms (SAC and TQC) to a laser light coupling task in a real experimental setting. By comparing the performance of RL agents against human experts, the authors demonstrate the potential of using RL to automate optics experiments.

**Audience:**

No

**Broader Impact Concerns:**

There are no ethical concerns regarding the broader impact of this paper.

**Claims And Evidence:**

Yes

**Requested Changes:**

1. Expanding the evaluation to include at least one additional real-world experiment that differs significantly from the current one to demonstrate generalizability of the design choices.

2. Testing more recent sample-efficient algorithms such as CrossQ [1], BRO [2], and TD-MPCs [3, 4] to strengthen the experimental comparison.

3. Consider submitting to venues more focused on applications of AI in experimental physics or optics, where the practical contributions would be more aligned with the journal's scope.

## References:
[1] Bhatt, Aditya, et al. "CrossQ: Batch Normalization in Deep Reinforcement Learning for Greater Sample Efficiency and Simplicity." *The Twelfth International Conference on Learning Representations*.

[2] Nauman, Michal, et al. "Bigger, regularized, optimistic: scaling for compute and sample efficient continuous control." *Advances in Neural Information Processing Systems* 37 (2025): 113038-113071.

[3] Hansen, Nicklas A., Hao Su, and Xiaolong Wang. "Temporal Difference Learning for Model Predictive Control." *International Conference on Machine Learning*. PMLR, 2022.

[4] Hansen, Nicklas, Hao Su, and Xiaolong Wang. "TD-MPC2: Scalable, Robust World Models for Continuous Control." *The Twelfth International Conference on Learning Representations*.

**Strengths And Weaknesses:**

## Strengths:
1. The paper presents interesting experimental results applying RL to a laser light coupling task, addressing practical challenges including partial observability and motor imprecision through careful design of the observation space and goal curriculum.
2. The reasons behind design choices (observation space, goal curriculum) are explained in detail and justified through ablation studies.
3. The comparison against human experts demonstrates the potential of RL systems for automating optics experiments.

## Weaknesses:
1. **Scope concern**: This paper does not propose modifications to existing RL algorithms or provide in-depth insights into RL methodology. It primarily uses established algorithms (SAC and TQC) as tools without advancing machine learning research, which, in my opinion, falls outside TMLR's scope as a machine learning research venue.
2. The evaluation is limited to a single task, which raises questions about the generalizability of the conclusions and design choices to other domains. This narrow focus further suggests the work may be better suited for a journal focused on experimental optics rather than TMLR.
3. The paper overlooks more recent advances in off-policy RL for continuous action spaces, such as CrossQ [1] and BRO [2], which could offer higher sample efficiency than SAC and TQC.
4. Since sample efficiency is a major challenge for real-world applications, it's unclear why only model-free methods were considered when model-based RL approaches such as TD-MPCs [3, 4] typically demonstrate stronger performance with limited data.

---

> ### Author Response · Authors · 2025-04-30
>
> We appreciate your feedback and understand that your main concern is the scope of the journal, specifically whether our manuscript aligns with the TMLR community's interests and expertise. Still, we believe that this is the case. While we don't provide any changes to the algorithm, we do apply known algorithms to a novel area. We see it as a feature of our work that we have applied standard algorithms without modification. Besides the development of algorithms, their  applicability in real-world scenarios is a crucial feedback for the development of algorithms and therefore for the TMLR community. Furthermore, our findings are not limited to the optics community, but of wider interest to any applied field.
>
> In addition, the transition from sim to real is often a major issue and learning directly on the real-world experiment still not common. We show success in this little-explored field and with this circumvent sim to real issues. We understand that this is a subjective matter and that our manuscript may only be relevant to a subset of the TMLR community. Notably, the other two reviewers did not raise any concerns about the audience.
>
> We are interested in the applicability of our findings to other scenarios, and have begun investigating other actuators and plan to extend our investigation to interferometric alignment. Given the time-consuming nature of these experiments, which typically take several months to complete, we cannot incorporate them within the revision period.  Besides changes in our hardware and experimental setup, we started comparing further algorithms, including those mentioned, in particular model-based RL algorithms to reduce training time.

---

### Review · Reviewer_6WCJ · 2025-04-23

**Summary Of Contributions:**

The paper investigates a suite of reinforcement learning methods for experimental control in optics. They perform their experiments in-situ using live actuators, and demonstrate that their method can outperform human experts in this domain. Some effort is applied to optimize the reward function and MDP set-up, and curriculum learning is used in order to improve performance on particularly difficult objectives.

**Audience:**

Yes

**Broader Impact Concerns:**

None.

**Claims And Evidence:**

Yes

**Requested Changes:**

If it is particularly easy, I would recommend extension to other set-ups (more actuators, different ambient conditions), but as this is difficult to perform for a journal revision, I will not require any such adjustments.

**Strengths And Weaknesses:**

I think the motivation for this line of work is quite strong as modeling environments accurately is indeed a major challenge. I am not familiar with the precise details of this setting, so I will refrain from commenting. It seems to me that there is much more that could be investigated in terms of experimental details, but the paper covers many bases.

In particular, I will highlight that I found the application of curriculum learning to be creative and quite persuasive in demonstrating its effectiveness.

The parametric form of the reward function still seems somewhat arbitrary, although the authors have put much effort into justifying it.

Overall, I believe this paper contains interesting points. It is difficult for me to compare it against prior work, which presents me from recommending it with full confidence (especially given my lack of familiarity). Nonetheless, I commend the authors for their effort.

---

> ### Author Response · Authors · 2025-04-30
>
> Thank you very much for your appreciative feedback.
>
> Regarding your requested changes: We are interested in testing our results on other systems and investigating the extent to which the training results are transferable. We have already acquired new motors and expect to test them in the next few months. Furthermore, we plan to expand our experiment to other applications such as the interferometric alignment of two beams. However, the amount of work required to set up and acquire reliable results exceeds our capabilities within the revision period and is beyond the scope of this publication.

---

### Author Response · Authors · 2025-04-30
**Log of Changes to the Paper**

Thank you for your comments. We would like to provide an overview of the changes we made in the newly uploaded version of the paper:

We added a  paragraph at the beginning of Section 2 (“Related Work”) to discuss differences of our setup to classic noisy environments in robotics.

We rephrased some explanations of the reset method at the end of the “Episodes and Resets” paragraph in Section 4 (“Our Method”).

We added the paragraph “Comparability of the virtual testbed and the real world experiment” at the end of Section 4 (“Our Method”)

Reformulated the caption of Table 3.

We further elaborated about means of reducing training time in paragraph 5 in Section 6 (“Summary and Outlook”).

---

### Decision · Action_Editor_zgnW · 2025-06-09

**Recommendation:** Accept with minor revision

**Additional Comments:**

I do agree with Reviewer sR4c that this paper has some limitations from the perspective of machine learning papers, and recommend the authors to make appropriate modifications based on the reviews. In particular, as Reviewer sR4c has suggested, if possible, please compare with more recent methods and/or explore extensions beyond a single task.

**Audience:**

Yes

**Audience Explanation:**

I believe that some individuals in TMLR's audience will be interested in knowing the findings of this paper. As the authors have clarified in the response, this paper has applied standard algorithms to a novel real-world scenario. I agree with the authors that this paper provides useful feedback for the applied standard algorithms.

Reviewer sR4c has some concerns about the scope of this paper, which I understand and partially agree with. My understanding is that, compared to a standard TMLR paper, the fraction of TMLR's audience who will be interested in the findings of this paper is expected to be smaller, but some individuals will be interested in it.

**Claims And Evidence:**

Yes

**Claims Explanation:**

To the best of my knowledge, the claims made in the submission are supported by accurate, convincing, and clear evidence. All three reviewers agree with this.